# The Role of MicroRNA in Graft-Versus-Host-Disease: A Review

**DOI:** 10.3390/genes14091796

**Published:** 2023-09-13

**Authors:** Martina Pitea, Filippo Antonio Canale, Gaetana Porto, Chiara Verduci, Giovanna Utano, Giorgia Policastro, Caterina Alati, Ludovica Santoro, Lucrezia Imbalzano, Massimo Martino

**Affiliations:** 1Stem Cell Transplantation and Cellular Therapies Unit (CTMO), Department of Hemato-Oncology and Radiotherapy Grande Ospedale Metropolitano “Bianchi-Malacrino-Morelli”, 89124 Reggio Calabria, Italy; filcan87@gmail.com (F.A.C.); porto.tania25@gmail.com (G.P.); verducichiara92@gmail.com (C.V.); giovanna.utano@ospedalerc.it (G.U.); giorgia.policastro00@gmail.com (G.P.); ludovicasantoro21@gmail.com (L.S.); lucrezia.imbalzano@ospedalerc.it (L.I.); dr.massimomartino@gmail.com (M.M.); 2Hematology Unit, Department of Hemato-Oncology and Radiotherapy Grande Ospedale Metropolitano “Bianchi-Melacrino-Morelli”, 89124 Reggio Calabria, Italy; caterina.alati@ospedalerc.it

**Keywords:** miRNAs, GVHD, stem cell transplantation

## Abstract

Allogeneic hematopoietic stem cell transplantation (allo-HSCT) is a clinically challenging modality for the treatment of many hematologic diseases such as leukemia, lymphoma, and myeloma. Graft-versus-host disease (GVHD) is a common complication after allo-HSCT and remains a major cause of morbidity and mortality, limiting the success of a potentially curative transplant. Several microRNAs (miRNAs) have recently been shown to impact the biology of GVHD. They are molecular regulators involved in numerous processes during T-cell development, homeostasis, and activation, and contribute to the pathological function of T-cells during GvHD. Here, we review the key role of miRNAs contributing to the GvHD; their detection might be an interesting possibility in the early diagnosis and monitoring of disease

## 1. Introduction

Allogeneic hematopoietic stem cell transplantation (allo-HSCT) is a potentially curative treatment for various hematological malignancies and non-malignant diseases. One of the most frequent complications of allo-HSCT is graft-versus-host disease (GVHD), a clinical syndrome in which the graft recognizes the transplant recipient as foreign. GVHD is the most serious complication of allogeneic hematopoietic cell transplantation; it can cause damage to organs and systems and can lead to death. The term graft-versus-host disease (GVHD) to define this disorder is due to Morton Simonsen and was first described in 1959 [1].

Billingham defines three crucial requirements for the development of GVHD, which are: (1) the graft must contain immunocompetent cells, (2) the host must possess allo-antigens that can be recognized as foreign by the immunocompetent cells in the graft, and (3) the host must be unable to mount an appropriate immune response against the graft [2]. Risk factors for GVHD include incompatible HLA match, advanced age of recipient/donor, multiparous female donor to male recipient, use of peripheral blood stem cell grafts rather than bone marrow, intensity of regimen conditioning, and CMV serological status [3]. GVHD has classically been divided into acute (aGVHD) or chronic (cGVHD) subtypes based on the time of onset before or after posttransplant day 100. Acute GVHD has several grades (grade 0–4), depending on the number and extent of organ involvement. Patients with grade 3 or 4 GVHD have a poor prognosis [4].

Thanks to the recently developed prophylactic strategies [5], incidences of aGvHD grades 2–4 decreased from 40% to 28% and the overall survival (OS) of patients experiencing GVHD has improved [6], but there is still a need for further progress. On the other hand, cGvHD remains the prevailing cause of non-relapse mortality (NRM) in patients surviving longer than two years after allo-HCT. New insight into the pathogenesis of GVHD could be the basis for the development of novel immunosuppressive therapies for the treatment of GvHD which are more effective and therefore able to decrease the administration of corticosteroids and lower the NRM of patients following allogeneic HSCT.

Currently, the diagnosis of aGVHD is based on clinical symptoms in one or more of the main target organs and biopsy results. So far, there is no validated diagnostic or predictive blood biomarker for aGVHD in clinical use. Thanks to innovative advances in sequencing techniques, it was discovered that the non-coding sequences of the human genome can be defined as functional molecules. In particular, non-coding RNAs (ncRNAs) control cellular responses by regulating gene expression, and previous studies have shown that these tiny molecules, particularly microRNAs (miRNAs), can influence allogeneic T cell responses in both animal and animal models in clinical trials [7].

In addition to the number of altered immune cell subsets, the balance of pro- and anti-inflammatory cytokines, chemokines, soluble cell receptors and proteins, miRNAs, extracellular vesicles (EVs), and activated immune biomarkers play a key role in both GVHD initiation and progression. GVHD-associated serum biomarkers, reflecting the underlying biological process of both aGVHD and cGVHD, have been shown to be useful not only in predicting the onset of GVHD before the clinical symptoms, but also in estimating its risk and predicting patient outcomes (Table 1) [8]. GVHD is the most feared disease after bone marrow transplantation, and it is also the most studied because nowadays there is no single mechanism that can identify its onset. For this reason, it is of fundamental importance to find a biomarker that can predict the onset or at least anticipate the symptoms, in order to be able to act to prevent it. The purpose of this review is to describe the existing knowledge and the state of the art in miRNAs regarding GVHD, distinguishing aGVHD and cGVHD, addressing the most relevant miRNAs studies conducted in various stages of GVHD, and also the recent investigations on miRNAs specifics associated with GVHD. It also aims to highlight the importance of miRNA investigations in the field of biomarker discovery and provide insight into the diagnosis and monitoring of a disease as difficult to prevent as GVHD.

**Table 1 genes-14-01796-t001:** Regulation miRNA in GVHD.

miRNA	aGVHD	cGVHD	Ref.
miRNA-155	**↑**		[9,10]
miRNA-146a	**↑**		[9,10]
miRNA-365a-3p		**↑**	[11]
miRNA-148a-3p miRNA-378a-3p		**↓** **↓**	[12,13] [12,14]
miRNA-29c-3p		**↓**	[15]

## 2. miRNA

Small non-coding miRNAs, which were originally described by Lee et al. [16], comprise a group of short regulatory molecules involved in all physiological processes. They play a crucial role in post-transcriptional gene expression regulation [17,18]. MiRNAs, which are roughly 19–22 nucleotides long, are partly similar to several mRNAs, and bonding with mRNA results in the degradation or downgrading of gene expression via numerous mechanisms, including translational repression, mRNA cleavage, and deadenylation [19]. 

MiRNAs are created via a multi-stage procedure: the initial transcript, pri-miRNA, is cleaved in the nucleus and then delivered to the cytoplasm as a precursor, pre-miRNA. In the final stages, pre-miRNAs undergo modification and the double strands separate into distinct single-stranded products. One of these strands partners with Argonaute (ARGO) family proteins to form the miRNA-induced silencing complex (miRISC), whilst the other is degraded (Figure 1) [20,21]. The miRISC has a crucial role in post-transcriptional gene regulation by facilitating the binding of the miRNA′s 5′ end to complementary sites on the 3′ mRNA and thus restricting translation. Additionally, it accelerates deadenylation and decapping of mRNA in the 5′-to-3′ direction, triggering mRNA decay [22,23]. Two pathways of regulation, namely repression of translation and downregulation of gene expression via mRNA degradation, are governed by miRNA to control gene expression [24]. It is widely documented in the literature, with specific evidence, that variations in miRNA levels are related to the pathogenesis of various diseases [25].

Thus, miRNAs regulate various cellular processes, including differentiation, proliferation, migration, and apoptosis. 

In just over two decades since the discovery of the first microRNA (miRNA), the field of miRNA biology has considerably expanded. Notable observations concerning the role of miRNAs in development and disease, especially cancer, have led to miRNAs becoming desired tools and targets for novel therapeutic interventions. Functional research has validated that miRNA dysregulation plays a causal role in many cancer cases, with miRNAs acting as either tumor suppressors or oncomiRs (oncogenes). There is potential in the preclinical development of miRNA mimics and antimiRs, which target miRNAs [26]. Currently, more than a thousand miRNAs have been identified, comprising 1% to 5% of all genes in the human genome. The activity mechanism of miRNAs focuses on binding to the 3′ untranslated region (UTR) of mRNAs with complementary seed sequences, which determines their destiny. Technical terms’ abbreviations will be defined when initially utilized. In regular hematopoiesis, miRNAs govern the differentiation, function, status, and self-renewal potential of HSCs. They also control apoptosis and the differentiation of myeloid and lymphoid progenitor cells [27]. Moreover, miRNAs manage the production of erythrocytes, platelets, and leukocytes, and the differentiation of B and T-lymphocytes. In malignant haematopoiesis, this function becomes altered, resulting in changes to the expression of specific miRNAs or clusters, which, in turn, disrupts the overall balance. Furthermore, recent studies have shown that miRNAs are present in biofluids. They remain protected from RNase-mediated degradation by being encapsulated in EVs or by binding to protective proteins, resulting in increased circulation of miRNAs in biofluids. This consequently leads to an extension of the time during which miRNAs remain in circulation [28].

## 3. miRNA as Biomarkers

Many studies conducted in the last decade have shown that approximately 50% of all genes are regulated by miRNAs, further underlining the importance of understanding their involvement in disease and being able to associate a specific miRNA with a specific disease [29].

The interest in these small miRNAs stems from the availability of potential diagnostic circulatory miRNAs [30] that could reduce the need for invasive methods, such as biopsies for the diagnosis of GVHD in allo-HSCT patients, and aid in disease monitoring or diagnosis. In the literature, it is reported that there is a link between miRNAs and different types of tumors, in particular the miRNAs circulating in plasma and serum are abundant and stable and this strengthens the hypothesis that the miRNAs of tumor origin are translocated in the blood and can act as tumor biomarkers or other diseases [31]. Circulating miRNAs can be detected in different types of biofluids by minimally invasive methods using relatively simple and accurate technologies. Furthermore, circulating miRNAs may offer advantages over protein-based biomarkers as they are of lower complexity, conserved in clinically relevant species, specifically expressed in different tissues or biological stages, and easily measurable using common laboratory techniques [27]. 

miRNAs were first discovered as biomarkers for cancer in 2008 when Lawrie et al. used them for the examination of diffuse large B-cell lymphoma in patient serum [15] and, since then, their potential use as biomarkers has been mentioned in the literature for numerous diseases. In particular, Lawrie’s group was the first to identify miRNA-155 as a potential biomarker because it was upregulated in the serum of patients compared to healthy subjects; however, the mechanism or which gene was linked to this upregulation was still unknown. 

The gold standard for miRNA quantification is quantitative reverse transcriptase polymerase chain reaction (RT-qPCR) [32]. This technique based on reverse transcription (RT) is the main technique used in molecular biology for the study and analysis of RNA, and in particular of miRNAs, with the advantage of having very high sensitivity and specificity rates [9]. This method is characterized by two phases; the first phase requires the binding of the miRNA molecules by the primers at the 3′ end to proceed with the reverse transcription of the stem loop. In the second step of the technique, real-time PCR is used to quantify the specific miRNAs that the operator wishes to study [33]. Another technique used is RT-based and poly(A)-based direct SYBR miRNA assays; this technique lower sensitivity [9]. The disadvantage of these techniques is that sometimes detection errors in the samples used can occur which can alter the quantitative reading of the sample, creating artifacts; furthermore, during the amplification steps, there is a high risk of contamination [32]. Following these discoveries, which deepened and expanded over the years, the attention of researchers has shifted to the study of GVHD, whose specific mechanism of action is still a mystery. Many advances have been made in the diagnostic field and for this reason the studies linking GVHD to miRNAs have increased, since finding a diagnostic system capable of preventing the symptoms of this disease remains a crucial point.

## 4. miRNA Involved in aGVHD

Although miRNA studies in relation to GVHD are still in their infancy, miRNA-155 was the first miRNA to be associated with aGVHD [28]. In 2013, Xiao’s group identified a group of plasma miRNAs as potential biomarkers for aGVHD. They drew their attention to 6 miRNAs (miRNA-423, miRNA-199a-3p, miRNA-93*, miRNA-377, miRNA-155, and miRNA-30a) that were significantly upregulated in the plasma of aGVHD patients compared to non-GVHD patients after allo-HSCT. In vivo experiments showed that miRNA-155 expression was upregulated in the T cells of mice with severe aGVHD after allo-HSCT and instead a reduction in miRNA-155 results in decreased aGVHD severity and prolonged survival in mice [34]. The researchers focused their attention on miRNA-155 because it is encoded within the integration cluster of B cells and is important for the regulation of acute inflammation and innate immunity. miR-NA-155 expression can be activated by inflammatory mediators, such as IFN-α/γ and TNF-α [35] (Figure 2), and Ceppi et al. [36] proposed that it functions as part of the negative feedback loop that controls inflammatory cytokine secretion by LPS-induced DC activation. Xiao’s research team identified 6 miRNAs, those mentioned at the beginning of this paragraph, that were present in the plasma of aGVHD patients involved in inflammation, tissue damage, regulation of cell proliferation, and tissue repair. In particular, miRNA-155 plays a key role in the regulation of inflammation and immune responses [37]. Indeed, it appears this miRNA is upregulated in the effector T cells of aGVHD mice and that miR-155 expression in lymphocytes is essential for lethal aGVHD in mice [28]. This result has been confirmed by other researchers over the years, for example Lin-Na Xie’s group in 2014 showed that miRNA-155 expression was upregulated in the T cells of mice developing aGVHD after allo-HSCT and the blockade of miRNA-155 expression indicated aGVHD severity, which decreased after allo-HSCT and survival was prolonged in mice [10]. This study highlighted that miRNA-155 level correlates with aGVHD severity; miR-155 expression was upregulated in patients with third and fourth stage aGVHD, i.e., severe stages. Furthermore, in patients with intestinal GVHD, the expression of miRNA-155 was particularly higher than that of the hepatic GVHD group. In 2016, the Atarod’s group started to investigate whether the expression of two miRNAs (miRNA-155 and miRNA-146a) could be used to predict aGVHD before the onset of the disease. This knowledge could make clinical interventions possible before the onset of the disease and, therefore, prevent the complications associated with aGVHD [38]. This study showed that the statistical interaction of miR-146a and miRNA-155 expression levels at day +28 post-transplant was significant in predicting aGVHD incidence, and investigation of the data revealed the conditional regulation mechanism wherein when both miRNAs were expressed at lower levels, there was a higher incidence of aGVHD. This observed statistical interaction is related to the “checkpoint” differential mechanism shown by Schulte et al. [39], in which by using various immune system stimulators, such as lipopolysaccharides (LPS), that trigger the expression of miRNA-146a [34] the authors showed that miRNA-146a was upregulated first, thus initiating the inflammatory response. This underscores the concept hypothesized that there is a time to onset of miRNA circulation. miRNA-155 expression was only altered when miRNA-146a levels of inflammatory tolerance were exceeded [12]. 

This may explain why higher miRNA-155 expression levels have been observed in both the serum and gut of patients at the time of aGVHD onset, which is the effector phase of the GVHD cycle [11]. Their results support the mice aGVHD study which showed that miRNA-146a expression levels were lower in severe aGVHD cases [40]. This may suggest that the overexpression of miRNA-146a prevents an inflammatory response and may have a protective role in innate immunity, the negative regulation of T-cells from donors by targeting TNF receptor associated factor 6 (TRAF6), and the subsequent TNF transcription inhibition [13]. The pathophysiology of GVHD is very complex and the study of this pathology is even more complex because most of the studies in the literature focus on animal models, which cannot fully mimic the pathophysiology of human GVHD. A clinical analysis of blood levels of specific miRNAs in patients with GVHD could be related to the development, diagnosis, and prognosis of GVHD.

In recent years, specifically in 2021, Crossland’s research team conducted a study by performing a comprehensive expression profile of miRNA-146a, miRNA-155, and miR-NA-155* expression in the target tissue and biofluids of patients with aGVHD and its post-HSCT expression [14]. Their findings are consistent with previously published observations that miRNA-155 and miRNA-146a play opposite roles during inflammation. In a transplant setting, as previously described, both miRNA-155 and miRNA-146a can be induced by LPS, which is a central component in triggering aGVHD pathology. These studies demonstrated that the expression of miRNA-155 and miRNA-146a is elevated in gastrointestinal aGVHD (gastrointestinal tract) compared with patients without gastrointestinal GVHD; expression was also elevated in the skin of patients with cutaneous GVHD compared with normal control skin (only the miRNA-155) and compared to pre-HSCT patients and patients without cutaneous GVHD (only the miRNA-146a). Furthermore, both miRNA-146a and miRNA-155 expression were elevated in the serum and urine of patients with generalized aGVHD at day 14 before the onset of symptomatic disease and also at aGVHD diagnosis. These results further support an important role obtained by the combination of miRNA-155 and miRNA-146a in aGVHD patients; however, the link between their involvement in generalized inflammation and specific organ pathophysiology requires further systemic investigations to confirm this correlation [41,42].

## 5. miRNA Involved in cGVHD

Chronic graft versus host disease (cGVHD) remains a leading cause of mortality and morbidity after allogeneic hematopoietic cell transplantation (allo-HCT). cGVHD is characterized by systemic inflammation, multiorgan fibrosis, and increased risk of infection [43]. Thus, cGVHD can be said to be a pleiotropic multiorgan syndrome involving tissue inflammation and the development of fibrosis, ultimately leading to organ dysfunction [44]. The role of miRNAs has also been studied in GVHD; however, most of these studies have been performed in aGVHD and our knowledge of miRNAs in cGVHD is generally limited and lacking due to the complexity of this disease, especially as it involves several organs simultaneously. There is a time window much studied by researchers, namely the one that exists between the appearance of symptoms and the beginning of the “circulation” of biomarkers. In fact, the primary objective of most studies is to study the time of onset of the disease, i.e., the window that allows for early diagnosis of cGVHD rather than an already advanced stage with very serious symptoms already evident, and it is precisely in this stage that miRNAs are of vital importance. Reikvam’s group, in 2022 [45], found that miRNA analysis was feasible in frozen serum samples and that the quality controls for the samples were satisfactory even after several years of freezing. They identified a heterogeneous miRNA pattern among allo-HSCT recipients, and a large amount of miRNA was detected in the majority of patients. Hence, their study highlighted the possibility of identifying large miRNA profiles among allo-HSCT recipients 1 year after transplantation. Initially looking for discriminatory miRNAs differentially expressed between patients who developed cGVHD and those who did not develop cGVHD, given the pleiotropic effects of miRNAs, the authors investigated miRNA profiles rather than individual miRNAs. Using bioinformatics tools to study miRNA profiles in allo-HSCT recipients, they identified different patient groups classified according to miRNA profiles. Most of the identified miRNAs were upregulated among cGVHD patients; indeed, the investigators focused their attention on miRNA-365a-3p as it was the most significantly altered miRNA among patients with and without cGVHD. This miRNA has aroused great interest as a previous study demonstrated that miR-365a is associated with nuclear factor κ-B (NF-κB) and the induced expression of the NF-κB subunit induces an upregulation of the promoter activity of the miRNA-365 [46]. NF-κB is involved in the stimulation of the expression of several inflammatory mediators and is thought to be crucial in the pathophysiology of GVHD [47]. The second identified miRNA, miR-148a-3p was also one of the miRNAs highly associated with cGVHD; it was also previously found to be significantly increased in GVHD patients [48]. Furthermore, miR-148a appears to activate Th1 cells, and the induced expression of miRNA-148a is believed to be crucial for chronic inflammation [49], concurrently, however, miRNA-148a downregulates the expression of the proapoptotic gene Bim, resulting in the survival of T cells that generate inflammation. Another miRNA associated with cGVHD in their study, miRNA-378a-3p, was previously found to be differentially expressed in patients with primary Sjogren’s syndrome compared with healthy controls. This chronic autoimmune syndrome has several common features with cGVHD, particularly symptoms from glands similar to cGVHD, and thus may indicate a potential role of miRNA-378a-3p in this process [50].

A very recent study by Lacina’s group in 2022 [51] sought to establish a panel of human EV miRNAs isolated from the plasma of a group of post-HSCT patients. They identified a potentially upregulated miRNA in cGVHD, miRNA-29c-3p, which has been extensively studied in various other diseases. It is reported in the literature in which it appears to be downregulated in acute lymphoblastic leukemia [52]. Furthermore, many studies have established that miRNA-29c-3p suppresses tumor cell proliferation and migration [53,54]. This study compared a group of only three patients who developed cGVHD with a group of four patients without cGVHD, with samples collected 3 months after transplantation. The number of patients studied was rather low as was the study time; therefore, the results presented should be treated with caution and are subject to validation in larger patient cohorts.

**Figure 1 genes-14-01796-f001:**
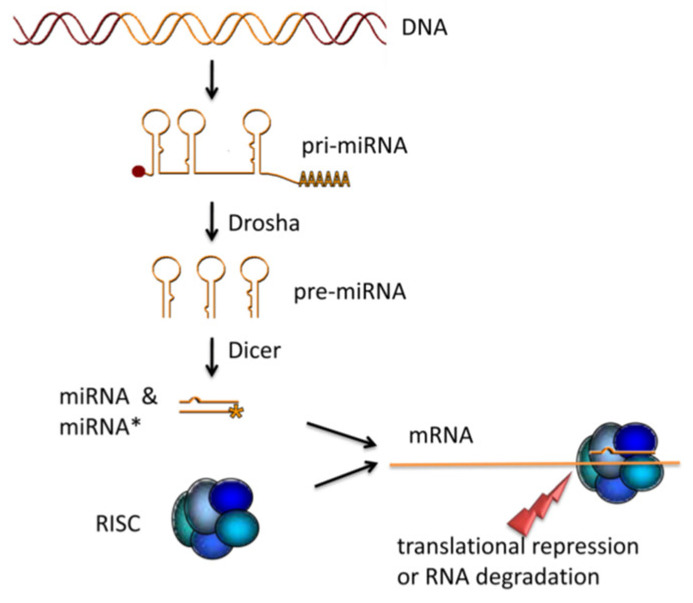
miRNA maturation. The primary transcript of miRNA (pri-miRNA) is processed by the endoribonuclease DROSHA. The consequently generated precursor miRNAs (pre-miRNAs) are transported from the nucleus to the cytoplasm and cut into 24 bp fragments by the DICER enzyme. After the double-stranded miRNA:miRNA* fragment is loaded into the RNA-induced silencing complex (RISC), the miRNA* is degraded. The RISC complex containing the mature miRNA binds to a target mRNA to inhibit translation by repression of translation or degradation of the mRNA [55].

There are many studies concerning cGVHD and they are not very detailed, this is because this disease involves different organs and it is difficult to identify the upstream mechanism. It is of fundamental importance, precisely for this reason, to identify a miRNA, or a cluster of miRNAs, or more generally a biomarker that can help clinical researchers with the diagnosis and monitoring of such a complex disease. So, although there are several pathways that have identified the potential miRNAs involved in cGVHD, they still need to continue to be studied.

## 6. Conclusions

Acute and chronic GVHD represent the main specific post-transplant complications affecting a patient’s life. The crucial point that distinguishes acute from chronic is the time of onset, followed by different clinical features, and above all the immunopathological mechanisms [56]. Acute GVHD (aGVHD) develops within the first 100 days posttransplant and affects the gastrointestinal tract, liver, skin, eyes, and oral mucosa as target organs [57]. Chronic GVHD (cGVHD) represents a delayed complication associated with transplantation and significantly reduced quality of life because it is a complicated multiorgan inflammatory syndrome [58]. Thus, timely diagnosis of GVHD is very important for patient outcomes as it is the main question that researchers seek to answer, and for this reason early biomarkers must be discovered for the prevention of transplant-associated morbid complications [59].

The discovery of miRNAs has triggered intense research to determine their role in normal and malignant haematopoiesis. The altered miRNA levels in cancer and healthy cells allow for the potential use of circulating miRNAs as diagnostic biomarkers. In this review, we wanted to highlight the advances reported in the literature regarding circulating miRNA levels in plasma or EVs, which represent a promising tool not only for aGVHD but also for the diagnosis, prognosis, and prediction of cGVHD.

In recent years, the attention on ncRNAs has increased, in particular circRNAs are being studied, a class of ncRNAs which have a closed circular structure and are about 500 ribonucleotides long, whose function is hypothesized to be the regulation of miRNAs. Studies have shown that circRNAs have a maximum half-life of 48 h, therefore circRNAs could be ideal biomarkers to prevent certain diseases. circRNAs perform an important function; sponge miRNAs, that is, bind miRNAs since they possess a specific binding site and they inhibit their action. circRNAs are widely expressed in mammals and participate in the regulation of physiological and pathological processes for various diseases, such as cancer, bone, and joint diseases [60]. Recently, many circRNA-related signaling pathways have been reported in autoimmune diseases, suggesting that circRNAs may serve as crucial immune regulators and potential biomarkers in combination with miRNAs. However, this is a recently discovered field and the mechanism is not yet known, so more studies are needed before defining this concept. However, although there have been many scientific advances on the study of miRNAs, further investigations of miRNA dysregulation in hematopoiesis may bring new strategies in miRNA-based approaches to improve outcomes for patients with hematologic cancer. Although miRNAs have many advantages, they also have limitations and for this reason they are not currently used as a diagnostic method. In order for these studies to be used in the clinical setting, it is necessary to broaden the field of study a large number of patients in order to be able to study all the facets of this vast world of miRNAs. In order to expand the number of patients, it is not 9only necessary to take into consideration the presence or absence of GVHD, acute or chronic, but also various factors such as age, gender, ethnicity, lifestyle, pre-treatment, history of diseases, and so on. Nonetheless, miRNAs represent the future of diagnostics and monitoring of many diseases and studying them and getting to know this world is essential for scientific progress.

**Figure 2 genes-14-01796-f002:**
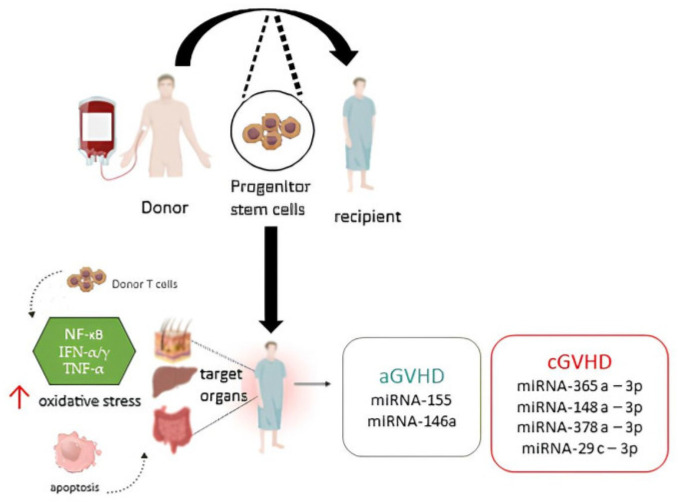
Schematic representative mechanism of action of the miRNAs studied in this review. Inspired by Sevcikova’s group [61]. The works studied in this review have highlighted that the miRNAs involved in aGVHD (green box) and cGVHD (red box) could be the biomarkers involving the NF-kB, INF- α/γ, and TNF-α pathways, which increase the levels of oxidative stress and cause apoptosis of cells in the target organs of aGVHD, and some of cGVHD.

## Data Availability

Not applicable.

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
