# Peer review of "The Role of MicroRNA in Graft-Versus-Host-Disease: A Review"

_genes, 2023, doi:10.3390/genes14091796_

Round 1

Reviewer 1 Report

This review is well-written. However, some issues should be addressed.

1. Please discuss the advantage of this review over the previously published one "doi: 10.7754/Clin.Lab.2021.210621".

2. I would suggest the authors include the description of circRNA and lncRNA in GVHD.

3. Please add a figure about the mechanisms of how miRNAs are involved in GVHD. 

Author Response

Dear Reviewer 1,

-1-

Ramzi and Shokrgozar's work on microRNAs studied in acute myeloid leukemia and graft versus host disease is a very well structured work. Our review aims to identify the most studied miRNAs in recent years in patients with aGVHD or cGVHD and to be able to identify a way to diagnose and prevent this disease whose mechanism is poorly understood. The message we would like to convey to the reader is that our review can be a guide for researchers studying the miRNAs involved in GVHD and from which they can base themselves to continue their scientific studies.

-2 and 3-

We have added a paragraph on circRNAs as suggested by you, we have also added in figure 2 the mechanism of action of miRNAs studied in this review.

Reviewer 2 Report

The manuscript reviewed miRNAs that are known to contribute to aGVHD/cGVHD. The topic is interesting and valuable in this field.

Below are some comments I have:

1. Please revise Table 1. Please make a table to briefly show published data including interactions between miRNAs and aGVHD/cGVHD, and by what potential mechanisms miRNAs affect aGVHD/cGVHD.

2. Please elaborate more on how miRNAs regulate aGVHD/cGVHD.

3. Manuscript writing can be improved - please conduct a thorough grammar and spelling check.

Moderate editing of English language required

Author Response

Dear Reviewer 2,

-1-

we have revised table 1 as suggested, we have added the references, but we have added the mechanism of action of the miRNAs studied in this review in figure 2.

-2-

We have added more details and more references on the miRNAs studied by learning about the target genes.

-3-

We have also done a thorough spell check of the English language.

Reviewer 3 Report

The review paper aims to identify the role of miRNA in Graft-Versus-Host-Disease, differentiating each of the miRNA occurring in acute or chronic disease, as an important step for diagnostic.

The review is clear and comprehensive with relevance in the field. The gap in the knowledge was identified. There is no similar review published recently, so the present one is relevant and of interest for the scientific community.

The review includes one original figure, fully described and easily to understand. The Table is coherent and depict very well the literature situation related to different expression of miRNA (down or up-regulated) identified in acute or chronic GVHD.

The statements and conclusions are coherent and supported by the listed citations. There are no self-citations listed in the references.

This is an important review study which tries to give a better view of the importance of miRNA in bone marrow/stem cells transplantation outcomes.

Minor type editing changes are required:

Line 86: please replace “microRNA” with miRNA (as this is the terminology used in the paper);

Line 149: please cut the space between 155 and closing bracket “(miR-155)”

Line 161: please put the reference “[34]” before the full stop;

Line 197: please put the reference “[44,45]” before the full stop;

Line 217: please correct the position of the full stop between “…cGVHD. However…”  

Author Response

Dear Reviewer 3,

thanks for your report! We performed an in-depth study on miRNAs in order to help researchers find an experimental way present in the literature that could help their work, which is so important in such a complicated disease.

We have changed the bugs you reported to us.

Reviewer 4 Report

The manuscript "The role of microRNA in Graft-versus-host-disease: a review", written by Pitea M, Canale FA, Porto G, Verduci C, Utano G, Policastro G, Alati C, Santoro L, Imbalzano L and Martino M. discusses the differential expression of several miRNAs in the cases of acute and chronic graft-versus-host disease (GVHD).

In the Introduction, appearance and symptoms of GVDH are descibed. The second paragraph presents the production and the roles of miRNAs, and in the following paragraphs miRNAs involved in acute and chronic type of disease are presented. In general, only several miRNA are described, and although some details involving symphtoms and roles of these miRNA are presented, for some of them these data are missing (such as target genes for miR146a and others.), as well as possible mechanisms, signaling pathway etc.. Similar topic has already been presented in a review article, with more details, such as in Vajari et al. J Cell Physiol 237, 3480-3495, 2022.

sentence reorganization: lines 28, 55, 104, 147, 191

Author Response

Dear Reviewer 4,

We have decided to report in our review the most studied miRNAs in the literature and those that in our opinion have proved to be the most relevant. We have added Figure 2 to illustrate the mechanism of action of the miRNAs studied in this review in order to facilitate understanding.

We also wanted to specify that Vajari's work is an experimental and very thorough article on the study of microRNA-146a and explains that it reduces the expression of MHC-II by targeting JAK/STAT signaling in dendritic cells after stem cell transplantation, our review aims to identify the miRNAs studied in the literature in recent years, in order to clarify and be able to identify a common pathway to diagnose and predict such an important disease as GVHD, both acute and severe.

Round 2

Reviewer 4 Report

The authors of the manuscript added a new scheme, but the main problem remained, and that is the existence of several review articles with similar topics. The example is article Vajari et al (2022), which is a review "Noncoding RNAs in diagnosis and prognosis of graftversus host disease (GVHD)".